# Detecting drought regulators using stochastic inference in Bayesian networks

**Aditya Lahiri**[1]*, **Lin Zhou**[2], **Ping He**[2,3], **Aniruddha Datta**[1,4]

**1** Department of Electrical and Computer Engineering, Texas A&M University, College Station, Texas, United States of America, **2** Department of Biochemistry and Biophysics, Texas A&M University, College Station, Texas, United States of America, **3** Institute for Plant Genomics and Biotechnology, Norman E. Borlaug Center, College Station, Texas, United States of America, **4** TEES-AgriLife Center for Bioinformatics and Genomic Systems Engineering (CBGSE), College Station, Texas, United States of America

* alahiri2@tamu.edu

**Data Availability Statement:** The dataset used in this study is publicly available at the NCBI GEO database with the accession ID of GSE42408 (https://www.ncbi.nlm.nih.gov/geo/query/acc.cgi?acc=GSE42408) All the code files and the relevant

## Abstract

Drought is a natural hazard that affects crops by inducing water stress. Water stress, induced by drought accounts for more loss in crop yield than all the other causes combined. With the increasing frequency and intensity of droughts worldwide, it is essential to develop drought-resistant crops to ensure food security. In this paper, we model multiple drought signaling pathways in Arabidopsis using Bayesian networks to identify potential regulators of drought-responsive reporter genes. Genetically intervening at these regulators can help develop drought-resistant crops. We create the Bayesian network model from the biological literature and determine its parameters from publicly available data. We conduct inference on this model using a stochastic simulation technique known as likelihood weighting to determine the best regulators of drought-responsive reporter genes. Our analysis reveals that activating *MYC2* or inhibiting *ATAF1* are the best single node intervention strategies to regulate the drought-responsive reporter genes. Additionally, we observe simultaneously activating *MYC2* and inhibiting *ATAF1* is a better strategy. The Bayesian network model indicated that *MYC2* and *ATAF1* are possible regulators of the drought response. Validation experiments showed that *ATAF1* negatively regulated the drought response. Thus intervening at *ATAF1* has the potential to create drought-resistant crops.

## Introduction

Drought is a natural hazard characterized by prolonged periods of dry conditions which can lead to economic, humanitarian, and ecological crises. In the context of agriculture, drought occurs when the amount of water available is not enough to sustain crops; such deficiency of water may arise from the lack of precipitation, soil water deficit, and reduced levels of ground or reservoir water [1, 2]. It is important to study the effect of droughts on agriculture as it is usually one of the first sectors to be impacted [3]. The United Nations Food and Agriculture organization reported that between 2005–2015 the agricultural sector of the developing countries suffered a loss of $ 29 Billion due to droughts [4]. In the United States, the state of California alone incurred a loss of 3.8 billion dollars from 2014–2016 due to the droughts which

subset of the dataset (GSE42408) supporting the conclusions of this manuscript are made available in the supporting files and at the following public GitHub repository: https://github.com/adilahiri/ Drought_Regulators.

**Funding:** This work was supported in part by the TEES-AgriLife Center for Bioinformatics and Genomic Systems Engineering (CBGSE) startup funds, the Texas A&M X-Grant Program, and in part by the National Science Foundation under Grant ECCS-1609236(to A. D.). This work was partially supported by the USDA NIFA Grant 2020-67013-31615 (to P.H). The funders had no role in study design, data collection and analysis, decision to publish, or preparation of the manuscript.

**Competing interests:** The authors have declared that no competing interests exist.

occurred from 2012 to 2016 [5]. Although the long term global drought trends have been a subject of debate, recent regional studies have shown an increasing trend of intensity and frequency of droughts across the Mediterranean, Western Africa, Central China, and Southwest and Central Plains of Western North America [6–10]. According to the special report published by the Intergovernmental Panel on Climate Change (IPCC) in 2018, human activities have contributed to global warming, and, at the current rate of warming, temperatures will rise by 1.5˚C between 2030 and 2052 [11]. This warming of the climate is projected to increase the frequency and intensity of droughts, especially in the southern African and Mediterranean regions [12]. Droughts are not caused by global warming alone; recent studies have shown that in the southwestern regions of the United States, droughts are expected to be more frequent and hotter due to structural changes in forested ecosystems and mass mortality of trees [13]. Along with being expensive events, droughts also threaten food security by affecting the global crop yield. With food security being a grand challenge due to a rising global population, frequent and more intense droughts in the future only serve to exacerbate this problem [14]. Thus, it is of paramount importance to develop crops that are robust against drought.

While the risk of imminent droughts has motivated the scientific communities' efforts in developing drought resilient plants, it has also led plants to develop and evolve their internal defense mechanisms to protect against droughts. Under drought conditions, plants can implement various strategies to conserve water to ensure their survival. For instance, plants can develop longer roots to search for water, shed their leaves early, slow their growth, or develop spines to conserve water in response to drought [15]. In addition to a plant's internal defense mechanism against drought, farmers have relied on traditional plant breeding methods such as selection and hybridization to combat drought. These methods have been successful in developing drought resistant plants in the past; however, progress has been slow due to the limited understanding of genetic and molecular interactions in the signaling pathways involved in the defense response of plants against drought [16]. Thus it is essential to develop a strong understanding of these signaling pathways. In this paper, we use Bayesian networks (BNs) to model the various drought signaling pathways of the model plant Arabidopsis. We use BNs as they allow us to combine biological pathway information along with experimental data, which is essential for developing a complete understanding of the interactions that take place inside a plant under drought conditions. We then perform inference using likelihood weighting in the BN model to identify targets in the pathways that regulate drought responsive genes. Genetically intervening (activating/inhibiting) at these target sites using methods such as CRISPR-Cas9 can help develop drought resistant plants [17].

## Plant defense mechanisms

Most living organisms can escape harsh environments by seeking refuge in favorable locations however, plants are immobile organisms and have to adapt to these conditions. If plants do not adapt to stressful conditions then their growth, development, yield, and seed quality may be hampered [18]. Plant stress can be categorized into two groups, biotic and abiotic. Biotic stress includes attacks on the plant by herbivores, bacteria, fungi, and other pathogens, whereas under abiotic stress the plant faces detrimental environmental conditions such as extreme temperatures, droughts, and mineral toxicity. Plants defend against such stress by activating complex networks of signaling pathways. These pathways are often activated with the help of small molecules such as $Ca^{2+}$, reactive oxygen species, nitrogen, or phytohormones such as ethylene, jasmonic acid, abscisic acid, and salicylic acid, which serve as biological stress sensors [19]. These pathway activators often initiate a protein phosphorylation cascade to directly target defensive proteins or transcription factors to regulate the stress responsive

genes [20]. Under stressed conditions, the natural metabolic homeostasis of plants is disrupted and, by activating the stress signaling pathways, plants achieve a new state of homeostasis; this process is commonly referred to as acclimation [21].

When a plant comes under drought conditions, it typically responds by implementing drought escape, avoidance, and tolerance strategies [22]. Drought escape strategies involve the plant developing high plasticity and completing its life cycle before the onset of drought, whereas under drought avoidance, the plant learns to maintain high water content in its tissues by increasing water uptake and reducing water loss [22–24]. Drought tolerant strategies are characterized by the plant developing traits such as epicuticular wax formation, osmotic adjustment, cellular elasticity, and protoplasmic resistance. These strategies allow the plant to survive in drought conditions with low tissue water content [24]. Plants do not deploy these defensive responses one at a time; instead, they implement a combination of these strategies to cope against drought [23]. Such a diverse range of defensive responses is achieved through the actions of Gene Regulatory Networks (GRNs) [24, 25]. GRNs are complex networks of genetic regulators called Transcription factors and their target genes; GRNs are directly responsible for altering the gene expression of plants when they receive environmental cues such as drought [26]. Due to these reasons, in this paper, we are interested in modeling the various GRNs, that are activated in plants in response to drought. Modeling these genetic interactions will help us establish a deep understanding of how plants deploy phenotypical defensive behavior through the actions of genes and transcription factors. Such a model will also help us identify the key regulators of drought response. The various GRNs involved in drought response in Arabidopsis are described in the following section.

## Drought signaling networks

In this paper, we build a BN model from several signaling pathways involved in the drought response of Arabidopsis. Since the plant's response to drought happens in a complex manner, it is necessary to build a comprehensive network model that can capture the multivariate and stochastic interactions taking place under drought conditions. Drought responses in plants are largely regulated by Abscisic acid (ABA) dependent and independent pathways [27]. ABA acts a sensor of drought in plants. Under drought conditions, the ABA levels increase rapidly in plants which allows them subsequently respond by closing their stomata and inducing drought responsive genes [28]. ABA regulates the expression of these genes through transcription factors in its drought signaling pathway. The basic-domain leucine zipper (*bZIP*) transcription factor and its subfamily of ABA-responsive element-binding protein/factor (*AREB*/*ABF*) constitute the primary transcription factors through which ABA regulates drought responsive genes [29, 30]. Under drought conditions, ABA induces *AREB1*(*ABF2*), *AREB2*(*ABF4*), *ABF1*, and *ABF3* from this transcription factor family in the vegetative tissues of Arabidopsis [31]. ABA and another plant phytohormone Jasmonic Acid (JA) regulate the expression of the drought responsive gene *RD22* in Arabidopsis via the transcription factors *MYB2* and *MYC2* [32, 33]. *MYB2* and *MYC2* act as a point of crosstalk between the ABA and JA signaling pathways. On the other hand, Dehydration-responsive element binding protein 1 (*DREB1*)/*CBF* (C-repeat binding factor) and *DREB2* transcription factor families operate independently of the ABA dependent pathway to regulate the drought responsive gene *RD29A*. This is achieved by the actions of transcription factors *DREB1A*(*CBF3*), *DREB1B*(*CBF1*), *DREB1C*(*CBF2*), and *DREB2A* [33, 34]. *DREB1A*, *DREB1B*, and *DREB1C* are negatively regulated by a transcription factor *MYB15* and positively regulated by another transcription factor, *ICE1* [35–37]. While *ICE1* negatively regulates *MYB15*, it is suppressed by transcription factors *HOS1* and upregulated by transcription factor *SIZ1* [38]. Among the various members of the DREB1 and *DREB2*

family, *DREB2A* and *DREB1D*(*CBF4*) play an interesting role in regulating drought response. Unlike the other *DREB* transcription factors discussed here, which function independently of the ABA pathway, *DREB2A* and *DREB1D* can be induced by the ABA pathway through the *ABRE* transcription factor family under drought conditions [33, 39, 40]. Therefore *DREB2A* and *DREB1D* serve as another point of crosstalk for both ABA dependent and independent pathways in regulating drought responsive genes. *DREB2A* was found to be further regulated by *DRIP1*. Singh et al. (2015) found that transgenic Arabidopsis overexpressing *DRIP1* delayed the expression of drought responsive genes regulated by *DREB2A* [33]. Downstream of the *DREB* and *ABRE* transcription factors is the drought responsive gene *RD29A* which is heavily regulated by these transcription factors [29, 40–42].

A recent study by Li et al. (2017) identified a drought stress-activated mitogen-activated protein (MAP) kinase cascade in cotton that regulates the expression of a drought responsive transcription factor *GhWRKY59*. *GhWRKY59* directly binds to the W-boxes of the transcription factor *GhDREB2* to regulate drought response in cotton [43]. We include this ABA independent pathway in our study of the drought regulatory network in Arabidopsis, where the MAP Kinase cascade is known to converge at the transcription factor *DREB2A*. In building our network model, we also study the *WRKY* transcription factor family which is traditionally associated with defense response against pathogens. However, many studies have now shown that *WRKY* transcription factor is involved in the defense response against drought [44–46]. The *WRKY* transcription factors *WRKY40*, *WRKY60*, *WRKY18* are induced by ABA to regulate the expression of *RD29A* [47]. *WRKY18*, *WRKY60* are known to positively regulate the expression of *RD29A*, whereas *WRKY40* inhibits *RD29A* and *WRKY60* [48]. Our previous paper on modeling the *WRKY* transcription factor in Arabidopsis under drought further confirmed these regulatory behaviors of the *WRKY* transcription factor family [49]. It should be noted that there is often crosstalk between ABA dependent and other independent pathways, we noted two instances of this earlier. Another instance of the crosstalk between the JA and ABA pathways was highlighted by Mintgen et al. (2014), where *WRKY60* from the ABA pathway suppresses the expression of *MYB2* in the JA pathway to regulate the drought responsive gene *RD22* [50]. Other than *RD22*, *MYB2* and *MYC2* also regulate the expression of another drought responsive gene *ERD1* [33]. According to a study by Ollas et al. (2016), *MYB2* and *MYC2* regulated the expression of *ERD1* through a cluster of transcription factors (*ANAC019*, *ANAC055*, and *ATAF1*) belonging to the *NAC* transcription factor family. *ERD1* was found to be further regulated by the transcription factor zinc finger homeodomain 1 (*ZFHD1*) and the gene *RD26* (*ANAC072*) in the ABA pathway [51]. In addition to the drought responsive genes *RD29A*, *ERD1*, and *RD22*, we also consider the gene *RD20* in our network model. *RD20* was found to be directly upregulated by the gene *RD26*(*ANAC072*) [51]. The biological interactions discussed above are summarized in Fig 1. In the next section, we create a Bayesian network model based on these signaling pathways to predict the best regulator(s) for the drought responsive genes (marked in blue in Fig 1).

## Materials and methods

### Bayesian network model

We observed in the previous section that plants deploy a diverse range of defense mechanisms to survive under drought conditions. These phenotypical defense responses are mediated through complex networks of signaling pathways at the genomic level. Biological signaling pathways have been successfully modeled using methods such as linear models, Boolean networks, probabilistic Boolean networks, Bayesian networks, and small molecule level models [52–57]. In order to develop a thorough understanding of these multivariate

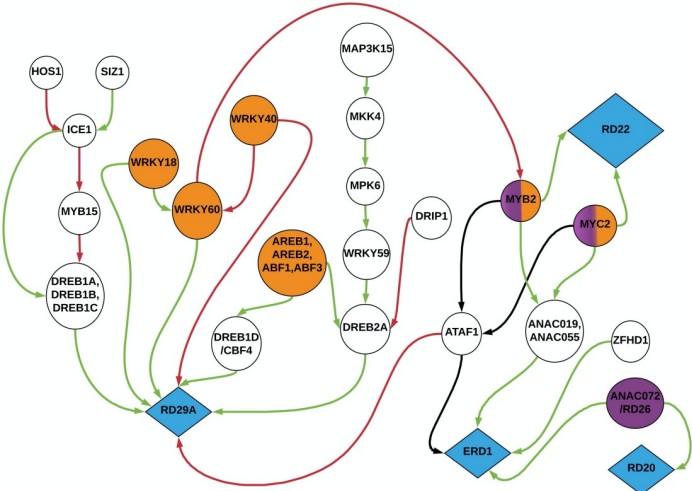

**Fig 1. Drought signaling pathways in Arabidopsis.** The orange circular nodes represent elements directly regulated by ABA whereas the purple nodes represent elements regulated by JA. The two nodes colored with a mix of orange and purple represent elements regulated by both JA and ABA pathways (Crosstalk). The blue diamonds represent drought responsive reporter genes. The plain circular nodes with no colors represent the transcription factors, genes and proteins involved in the regulation of drought responsive reporter genes in an ABA independent manner. The green and red arrows represent positive and negative regulation. The arrows going into and out of *ATAF1* are marked black to indicate that the nature of regulation is not known at this time.

and stochastic interactions, we create a BN model of the drought signaling pathways. Unlike some modeling techniques which are solely driven by data, a BN model allows us to integrate pathway information in the form of prior knowledge along with experimental data [58]. BNs are directed acyclic graphs that represent the causal probabilistic relationships among a set of random variables and provide the conditional decomposition of the joint probability distribution of these random variables [59, 60]. Thus BNs serve as an ideal modeling paradigm to study the drought signaling pathways [58]. In this paper, our objective is to create a BN model of the drought signaling pathways outlined in Fig 1 and use this model to determine which transcription factor, protein or gene is the best regulator of drought responsive reporter genes (blue diamonds in Fig 1). The predictions made by the model can help us identify potential targets for genetic intervention techniques like CRISPR-Cas9 to create drought resistant crops.

Fig 2 represents the BN model of the signaling pathways shown in Fig 1. Every node (circle) in the network represents a gene, protein, or transcription factor in the drought signaling pathway. The black arrows or edges connecting the nodes represent the causal biological relationships we discussed in the previous section. We assume each of the nodes are binary random variables that can assume 1 for activation and 0 for inhibition. Since the nodes are random variables, associated with each of them is a parameter $\theta$ which describes the local marginal or conditional probability distribution for that node. For instance, the conditional probability parameter associated with the node representing *MKK4* is given by $\theta_{MKK4|MAP3K15}$. This parameter represents the activation or inhibition probability of the node representing *MKK4* conditioned on the state of the node representing *MAP3K15*. Similarly, for the node representing the transcription factor *ICE1*, the local conditional probability distribution is given by $\theta_{ICE1|HOS1,SIZ1}$. Henceforth, we will refer to local conditional or marginal probability distribution as just local probability distributions (LPD). We learn these LPDs from experimental biological data; once these LPDs are learned, the BN model is complete and can be used for

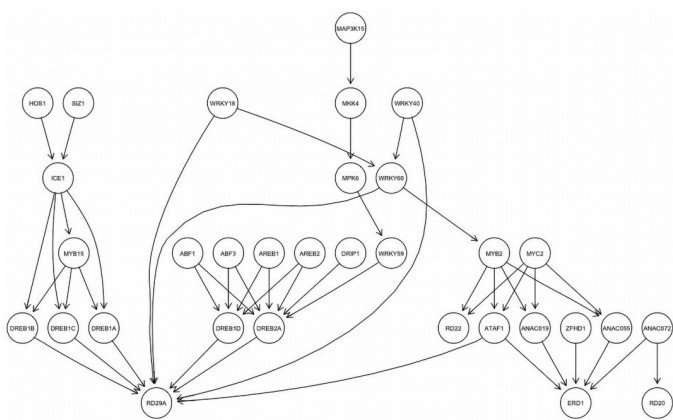

**Fig 2. Bayesian network model of drought signaling pathway.** Every circular node represents a biological element in the drought signaling pathway. Every edge or black arrow represents the causal biological relationship between the nodes. Associated with every node is a $\theta$ parameter that represents the local probability distribution of the node.

carrying out inference simulations to determine the best modulator for the drought responsive genes.

## Parameter estimation in Bayesian networks

BNs consist of two major components: a directed acyclic graph (DAG) and a set of local probability distributions. The DAG can be learned from data or constructed from domain knowledge. Learning BNs from data, also known as structure learning in the literature, is an NP-Hard problem and requires us to choose a DAG from several candidate DAGs [61]. This is not very practical as we observed in in the prior sections that pathway interactions are well defined, and there can only be a single DAG representing them. Furthermore, in the context of Arabidopsis under drought, we are limited by the sizes of publicly available datasets. These datasets are not large enough to construct a reliable DAG, so we elected to create the BN model in Fig 2 using pathway information from the existing biological literature. While a DAG can be learned either using data or from domain knowledge, the local probability distributions associated with the DAG have to be estimated from experimental data. There are several ways to estimate the local probability distribution in a BN model. Typically, either a frequentist approach such as a Maximum Likelihood Estimate (MLE) or a Bayesian approach is employed. Though methods such as MLE are simple and provide a point estimate, they are only driven by data and do not take any relevant prior information into account [62]. On the other hand, a Bayesian approach provides us with the posterior distribution, which is driven by both data in the form of likelihood and relevant information in the form of a prior distribution. However, the Bayesian approach has two significant drawbacks. The first one is computing the normalizing constant or the probability of data (evidence) [63]. The normalizing constant very rarely has a closed form solution and hence can be computationally expensive to determine. The second drawback pertains to the choice of a prior distribution. Since the choice of the prior distribution is subjective and there exists no established method to select one, different choices of prior distribution will lead to different results [64]. Nonetheless, the Bayesian approach is logically rigorous and unlike frequentist approaches, once the prior distribution is established the Bayesian approach follows deductive logic. In this paper, we use a Bayesian approach to estimate the local probability distributions for the BN model outlined in Fig 2. We assumed that the nodes are binary random variables, which implies that for any node **X** in the

BN, $\mathbf{X}$ = 1 (success) when the node is activated and $\mathbf{X}$ = 0 (failure) when the node is inhibited. Then for a single observation for any node $\mathbf{X}$ in the BN be modeled as a Bernoulli random variable.

Let us suppose that we have a BN model with $\mathbf{N}$ nodes. Then the probability with which any node $\mathbf{X}$ attains a state of 1 is given by $\theta_X$. Thus if we make $n$ (>0) independent and identically distributed observations (i.i.d) observations for each node in the BN, and if for a given node $\mathbf{X}$, we observe $k$ instances when the node attains a state of 1, then the likelihood for node $\mathbf{X}$ is given by:

$$P(X|P_a(X), \theta_X) \sim Binomial(n, \theta_X) \tag{1}$$

$$Binomial(n, \theta_X) = \frac{n!}{k!(n-k)!} \theta_X^k (1 - \theta_X)^{n-k} \tag{2}$$

$P_a(X)$ in Eq (1) refers to the parents, if any, of node $\mathbf{X}$. Since we are using a Bayesian approach to estimate the LPD of Node $\mathbf{X}$, we need to select a prior distribution on the node $\mathbf{X}$. Considering the computational complexity required in calculating the normalizing constant, and since the likelihood function associated with our model follows a binomial distribution by design, we assume the prior distribution on $\theta_X$ to follow a Beta distribution. Since the Beta and Binomial distributions belong to conjugate families, we know that the posterior distribution of $\theta_X$ will also follow a Beta distribution [65]. This is formulated as follows:

$$\theta_X \sim Beta(\alpha_X, \beta_X) \tag{3}$$

$$P(\theta_X|X) \sim Beta(\alpha_X', \beta_X') \tag{4}$$

where $\alpha_X' = \alpha_X + \text{k}$ and $\beta_X' = \beta_X + (\text{n} - \text{k})$.

In Eq (3), $\alpha_X$ and $\beta_X$ represent the shape parameters of the Beta distribution, and in Eq (4) these parameters get updated for the posterior distribution on $\theta_X$. We assume $\alpha_X = 1$ and $\beta_X = 1$ for our calculations as the Beta(1,1) distribution corresponds to the standard uniform distribution over the interval [0, 1] [66]. Setting the prior distribution to the standard uniform distribution guarantees that we have no information regarding the prior distribution of $\theta_X$. We chose the Beta(1,1) distribution as our prior because we do not have any domain knowledge information regarding the prior distribution of every node in the BN model. If we had such information regarding the prior distribution, they could be incorporated into this model. However, it is to be noted that choosing a different prior distribution may not allow us to reach a closed form solution for the posterior distribution on $\theta_X$. Since the result we get in Eq (4) is a distribution and not a point estimate like what we would have obtained had we used a frequentist approach, we approximate the values for $\theta_X$ with the expected value of the posterior distribution. We do this approximation for the posterior distributions estimated at every node in the BN. This approximation for the node $\mathbf{X}$ has been presented in Eq (5).

$$\theta_X \simeq E[\theta_X|X] = \frac{\alpha_X'}{\alpha_X' + \beta_X'} \tag{5}$$

Once these parameters are learned the BN is complete as we have both the DAG and the set of conditional probabilities. In the next section, we study the effect on drought responsive genes for intervening (activating/ inhibiting) at various nodes, then summarize our findings in the Results section.

## Sampling based inference in Bayesian networks

In this section, we are interested in using the BN model in determining which nodes are the best regulators of the drought responsive reporter genes *RD29A*, *RD20*, *RD22*, and *ERD1*. Specifically, we want to study the effect on the reporter genes of intervening at the non-reporter genes. In other words, we will fix the state of every non-reporter gene node one at a time to either 0 or 1, and observe how this action (intervention) affects the LPDs for the nodes representing the drought responsive reporter genes. This kind of simulation in BNs is known as inference. Inference techniques are categorized as either exact or approximate. Exact inference techniques such as Enumeration, Variable Elimination, and Pearl's Message Passing Algorithm are particularly efficient in polytrees or singly connected networks. One such application of exact inference was demonstrated by Vundavilli et al. to find significant nodes in the breast cancer signaling pathway [67]. Ideally, we would like to use an exact inference technique to calculate the LPDs in our BN model. However, exact inference techniques will be computationally expensive to implement as our network is multiply connected, i.e., there are at least two nodes in our BN model connected by more than one path. For instance, we can see that the nodes *DREB1A* and *ICE1* are directly connected and are also connected through *MYB15*, hence making our BN model multiply connected. While exact inference algorithms work in polynomial time in polytrees, it has been shown to be NP-Hard in more generalized BNs, hence implementing them in multiply connected networks may not be practical [68]. Therefore, the size and structure of the BN govern our choice of inference techniques. This is the reason why, for determining the regulators of drought responsive reporter genes, we employ an approximate inference technique known as likelihood weighting.

Likelihood Weighting (LW) is an approximate inference technique based on stochastic simulations. Inference techniques based on stochastic simulations usually involve drawing samples from a sampling distribution, calculating an approximate posterior probability based on the samples, and then showing that the posterior probability converges to the actual probability [69]. In the context of our model, the sampling distribution will be specified by the BN in the form of LPDs. Unlike exact inference techniques, LW is generally insensitive to the network topology, however, convergence in estimating the posterior probabilities can be slow if they are close to 0 or 1 [70]. We will now describe the mathematical formulation for LW.

Consider a BN consisting of N nodes such that the DAG follows a topological ordering of $\{X_1, X_2, .., X_N\}$. Suppose we make an observation on the node $X_E$ in the BN, we will refer to $X_E$ as the evidence node. Now suppose our objective is to find the effects of this observation on another node $X_Q$, known as the query node in the BN. Specifically, we want to estimate the posterior probability $Pr(X_Q = x_q | X_E = x_e)$, where '$x_q$' and '$x_e$' are some instantiation of nodes $X_Q$ and $X_E$. At this step we begin performing LW by drawing M samples from the BN for every node except for the evidence node $X_E$, in topological order. The generated dataset ($\xi$) will be a matrix with M rows and N columns, where each row represents an N-dimensional sample (datapoint) and columns represent nodes in the BN. Thus after the first iteration of the sample generation process, the datapoint will be of the form $\xi^{(1)} = \{x_1^{(i=1)}, x_2^{(i=1)}, \ldots, x_e^{(i=1)}, \ldots, x_N^{(i=1)}\}$. We will repeat this process M-1 more times to obtain M such samples, thus that dataset will be of the form $\xi = \xi^{\{i=1,2,..,M\}} = \{x_1^{(i)}, x_2^{(i)}, \ldots, x_e, .., x_N^{(i)}\}$. It should be noted that $x_e$, does not change across the M samples. This is because $X_E$ is the evidence variable that has been observed and fixed. The samples for the rest of the non-evidence nodes are generated according to the LPDs associated with those nodes. For example we draw a sample $x_1$ for root node $X_1$ according to $Pr(X_1)$. Similarly we draw a sample $x_2$ for the node $X_2$ according to $Pr(X_2 | X_1 = x_1)$ and so on. It should be noted that all the children of node $X_E$ have a fixed instantiation

for $X_E$, that is $x_e$. We then approximate $Pr(X_Q = x_q | X_E = x_e)$ as follows:

$$Pr(X_Q = x_q | X_E = x_e) \simeq$$

$$\lim_{M \to \infty} \frac{\sum_i^M \mathbb{1}[x_q^{(i)} = x_q] Pr(X_E = x_e | (P_a(X_E))^{(i)})}{\sum_i^M Pr(X_E = x_e | (P_a(X_E))^{(i)})} \tag{6}$$

The proof for Eq (6) is not trivial and is presented in a paper by Menon [71]. A psudo code for estimating the conditional probabilities using LW is presented in algorithm 1. We will now demonstrate LW on an example BN.

**Algorithm 1**: Psudo Code for likelihood weighting in Bayesian Networks

```
Input:
  1: BN: The Bayesian Network
  2: Q: The Query Variable, Let Q = q, that is node Q is instantiated
to some value of interest q.
  3: E: The Evidence variable. Let E = e, that is node E is instanti-
ated to some observed value e.
  4: M: Number of Samples.
Output: Probability: Estimate of P(Q = q|E = e)
  5: Initialization: X₁,X₂,□,Xₙ Topological Ordering of BN
    Sampled_Data = {} {}, M by N matrix to store sampled data
    Weight = {1,...,1}, an array of size M, consisting of weights
    with values initialized to 1.
    Counts[k] = 0, where k ∈ domain of Q
  6: while iter = 1 to M do
  7:   for each node X in BN in topological order do
  8:     if X = Xᵢ is in E then
  9:       Sampled_Data[iter][Xᵢ] = x, where x is the value of Xᵢ
 10:       Weight[iter] = Weight[iter] * P(Xᵢ = x | Pₐ (Xᵢ))
 11:     else
 12:       Sampled_Data[iter][Xᵢ] = Generate random sample from P(Xᵢ =
x|Pₐ(Xᵢ))
 13:     end if
 14:   end for
 15:   iter = iter+1
 16: end while
 17: k = List of row indices in Sampled_Data where Q = q
 18: Probability = Sum (Weights [k])/ Sum(Weights)
 19: return Probability
```

Fig 3 describes an example BN consisting of four genes A,B,C, and D. We consider the nodes representing the genes as binary random variables, which can take on the values of 1 for activation and 0 for inhibition. The LPDs for this example BN are already estimated and are presented in Fig 3. For the purpose of this example, we assume that Gene A positively regulates gene B, while it negatively regulates gene C. Gene D is upregulated by gene B, while gene C downregulates it. These effects are reflected in the LPDs for each node. Now suppose, we are interested in gene D being positively regulated, and we decide to intervene at Gene B and set it to 1. Therefore, node B = 1 serves as the evidence variable, and let us consider node D as the query variable. Then we are interested in finding the probability P(D|B = 1) using LW.

In order to estimate this probability, we will need to query the BN and generate samples first. We use the topological ordering of{A,B,C,D}, another valid ordering is {A,C,B,D}. The sample generation process is described in the following steps:

1. Set the weight variable 'W$_i$' to 1. W$_{(i)}$ = 1

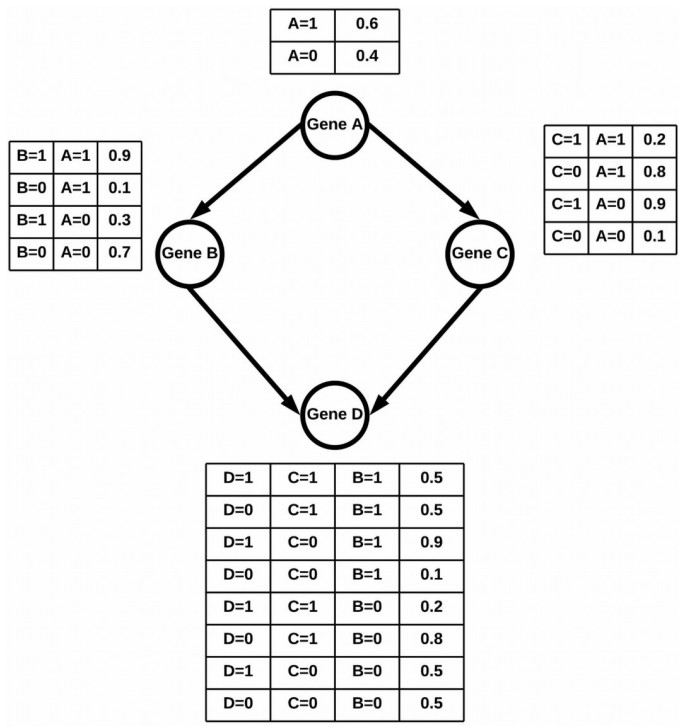

**Fig 3. Example BN with LPDs.** Gene A positively regulates Gene B and negatively regulates Gene C. Gene B positively regulates Gene D and Gene C negatively regulates Gene D.

2. The matrix Sampled_Data[iter][$X_i$] = is empty. This matrix will store the value of nodes A, B,C,D.

3. We start topologically at node A. Since A is not an evidence node, we sample it according to its LPD, specifically P(A). Assume this sample results in A = 1.

4. We now move on to node B. Since B is an evidence node, we do not sample it. We update, $W_{(i)}$ = 1. P(B = 1|A = 1) = 1. (0.9) = 0.9.

5. We now go to node C. Since C is not an evidence node, we sample it according to its LPD, specifically P(C|A = 1). Let us assume the result of this process is C = 0.

6. We now sample node D with its LPD of P(D|B = 1,C = 0). Assume that this results in D = 1.

7. The sample generated is (A = 1,B = 1,C = 0,D = 1) with $W_{(i = 1)}$ = 0.9. Thus Sampled_Data [1][All Columns] = [1, 1, 0, 0]

8. We repeat steps 1–7, M-1 more times to obtain a total of M samples.

9. We can then calculate P(D|B = 1) as follows:

$$P(D = 1|B = 1) = \frac{\sum_{i=1}^{M} W_i \mathbb{1}[D_{(i)} = 1]}{\sum_{i=1}^{M} W_i}$$

$$P(D = 0|B = 1) = \frac{\sum_{i=1}^{M} W_i \mathbb{1}[D_{(i)} = 0]}{\sum_{i=1}^{M} W_i}$$

**Table 1. Sample data from example Bayesian network.**

| index | A | B | C | D | Weight($W_i$) |
|---|---|---|---|---|---|
| 1 | 1 | 1 | 0 | 1 | 0.9 |
| 2 | 0 | 1 | 1 | 0 | 0.3 |
| 3 | 1 | 1 | 1 | 1 | 0.9 |
| 4 | 1 | 1 | 0 | 0 | 0.9 |
| 5 | 0 | 1 | 0 | 1 | 0.3 |

Therefore for M = 5, if we generated sample, it would result in a 5 by 4 matrix (Sampled_Data [iter],[$X_i$]). Table 1 shows this matrix with an extra column for weights belonging to each sample. From the samples and weights in Table 1, we can now estimate P(D = 1|B = 1) and P(D = 0|B = 1) as follows:

$$
\begin{aligned}
P(D = 1|B = 1) \quad &= \frac{\sum_{i=1}^{5} W_i \mathbb{1}[D_{(i)} = 1]}{\sum_{i=1}^{5} W_i} \\
&= \frac{W_1 * 1 + W_2 * 0 + W_3 * 1 + W_4 * 0 + W_5 * 1}{W_1 + W_2 + W_3 + W_4 + W_5} \\
&= \frac{0.9 * 1 + 0.3 * 0 + 0.9 * 1 + 0.9 * 0 + 0.3 * 1}{0.9 + 0.3 + 0.9 + 0.9 + 0.3} \\
&= \frac{2.1}{3.3} \\
&= 0.636364
\end{aligned}
$$

$$
\begin{aligned}
P(D = 0|B = 1) \quad &= \frac{\sum_{i=1}^{5} W_i \mathbb{1}[D_{(i)} = 0]}{\sum_{i=1}^{5} W_i} \\
&= \frac{W_1 * 0 + W_2 * 1 + W_3 * 0 + W_4 * 1 + W_5 * 0}{W_1 + W_2 + W_3 + W_4 + W_5} \\
&= \frac{0.9 * 0 + 0.3 * 1 + 0.9 * 0 + 0.9 * 1 + 0.3 * 0}{0.9 + 0.3 + 0.9 + 0.9 + 0.3} \\
&= \frac{1.2}{3.3} \\
&= 0.363636
\end{aligned}
$$

## Dataset and simulation

To estimate the LPDs for the nodes in the BN model, we needed gene expression data (e.g., microarray, RNA-Seq, eQTL, etc.) for Arabidopsis under drought conditions. We searched the NCBI GEO database and selected the dataset GSE42408 [72, 73]. We chose this dataset as it had gene expression data for the genes of interest in our BN model from 104 recombinant inbred lines of Arabidopsis under drought conditions. Furthermore, this dataset had the most number of data points per gene compared to other datasets found during the search of the NCBI GEO database, which also led to its selection for our analysis. This dataset contains 104 eQTL (expression quantitative trait loci) data points for Arabidopsis under drought conditions. The data for each node is normalized using min-max feature scaling. We further compute the normalized means for each node and use it as a threshold for binarizing the data. Additional details on the normalization and binarization process can be found in the R scripts

provided in the supporting information section. The processed data was then used to learn the LPDs for each node and perform inference using LW. We chose a sample size (M) of 600,000 in the LW algorithm to ensure convergence in estimating the conditional probabilities. The model building and all the associated data processing tasks were completed using the R programming language [74]. The Bnlearn package was used to perform inference using LW [75]. All the code and data files are also made available publicly at the following GitHub repository: https://github.com/adilahiri/Drought_Regulators.

## Results

Fig 4 displays the dataset GSE42408 after it was normalized and binarized. Each bar in Fig 4 represents the inhibition and activation counts for each node in the BN. We use the Bayesian approach as discussed in section 3.1, with Beta (1,1) as the prior distribution for each node to estimate the LPDs. For the inference analysis, the query nodes were the drought responsive reporter genes *RD29A*, *RD20*, *RD22*, and *ERD1*. We were interested in the activation of *ERD1* and the inhibition of *RD29A*, *RD20*, and *RD22*. Though all these reporter genes have been shown to confer drought resistant characteristics, they also impart undesirable traits such as sterility, reduced seed yield, and dwarfing [51]. Thus activating all of them is not optimal, hence for our analysis, we are interested in finding a single node which upon intervention would increase the chances of the reporter gene *ERD1* being activated and the reporter genes *RD29A*, *RD20*, and *RD22* being inhibited. Since the LW yields a probability for the status of every drought reporter node based on performing an intervention at an evidence node, we establish a composite scoring metric defined in Eq (7) below.

$$Score(Evidence = \{0, 1\}) =$$
$$Pr(RD29A = 0|Evidence = \{0, 1\})$$
$$Pr(RD22 = 0|Evidence = \{0, 1\}) \tag{7}$$
$$Pr(RD20 = 0|Evidence = \{0, 1\})$$
$$Pr(ERD1 = 1|Evidence = \{0, 1\}).$$

This metric multiplies the conditional probability for all the drought responsive reporter genes into a single number which is easy to interpret. A high score represents a suitable

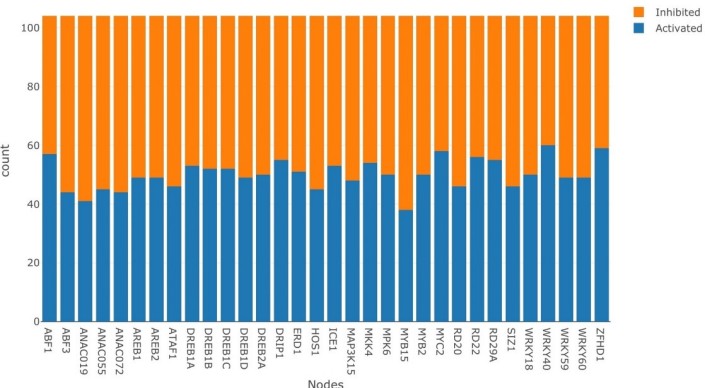

**Fig 4. Activation vs inhibition plot.** This figure represents the data after it has been normalized and then binarized. There are a total of 104 data points per node. The blue part of each bar represents activation counts whereas the orange part represents the inhibition counts.

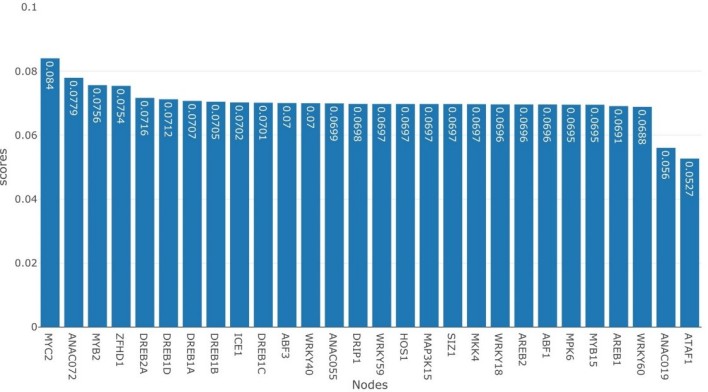

**Fig 5. Activation scores for non-reporter gene nodes.** Associated with each node is a blue bar which represents the score for activating that node.

candidate for intervention. Figs 5 and 6, we present the score for intervening at each of the non-reporter nodes one at a time in the BN. The non-reporter nodes are activated in Fig 5, whereas in Fig 6, they are inhibited. From Fig 5, it is clear that when *MYC2* is activated, it results in the highest score, whereas *ANAC072* and *ZFHD1* have the second and third highest scores, respectively. On the other hand, in Fig 6, *ATAF1* has the highest score for inhibition, followed by *ANAC019*. Our analysis shows that activating *MYC2* or inhibiting *ATAF1* maximizes the scores under single node intervention. Thus these are the best strategies to activate *ERD1* and inhibit *RD29A*, *RD20*, and *RD22*. We observe that the score for *MYC2* is the lowest when it is inhibited (Fig 6) and the score for *ATAF1* is lowest when it is activated (Fig 5), this makes logical sense for the analysis.

The above results from the single node intervention analysis motivated us to study effects on the drought reporter genes when we simultaneously intervened at *MYC2* and *ATAF1*. In Fig 7, we present the score of simultaneously activating *MYC2* and inhibiting *ATAF1*. Upon comparing this score to the individual scores of activating *MYC2* and inhibiting *ATAF1*, we notice that the score for the combined intervention is slightly higher, indicating the synergistic

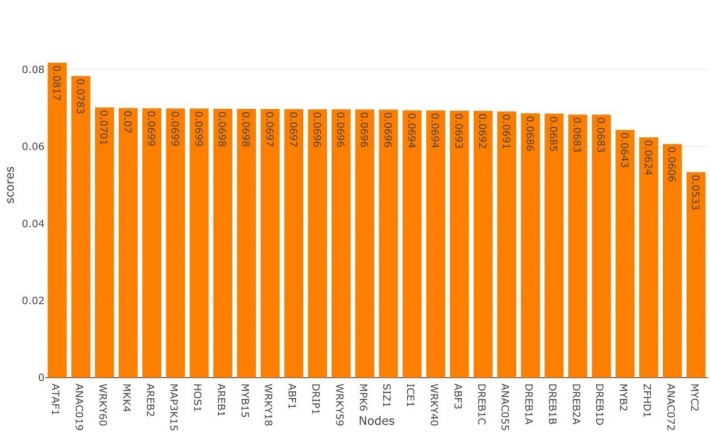

**Fig 6. Inhibition scores for non-reporter gene nodes.** Associated with each node is an orange bar which represents the score for activating that node.

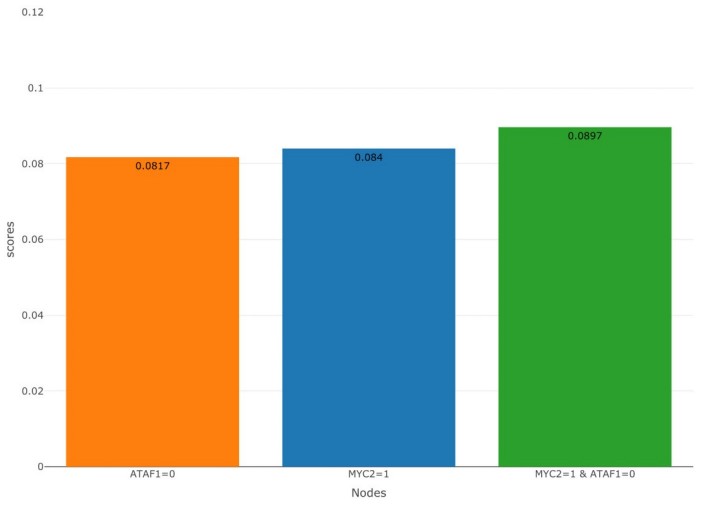

**Fig 7. Comparing the scores of multi-node and single node intervention under optimal response case.**
Simultaneous (multi-node) intervention on *MYC2* and *ATAF1* has a slightly higher score than single node intervention.

effect of intervening strategically at the two nodes. Furthermore, both *MYC2* and *ATAF1* are established regulators of the drought response [76, 77]. *MYC2* is known to be a positive regulator of the drought responsive reporter genes *RD20,RD22*, and *ERD1* [78–80]. A study found *MYC2* to have no significant regulatory effect on *RD29A* in Arabidopsis [81]. In contrast to the positive drought regulatory nature of *MYC2*, *ATAF1* is known to negatively regulate the expression of *RD29A* and *RD22* [82]. The regulatory effects of *ATAF1* on *RD20* and *ERD1* are not yet known. Due to *MYC2* being a positive regulator for most of the drought responsive reporter genes and *ATAF1* being a negative regulator for two of the drought responsive reporter genes, it is biologically consistent for them to be the best regulators under activation and inhibition, respectively.

## Experimental validation

To validate the conclusions from the Bayesian network model, we isolated Arabidopsis *ataf1* (SALK_057618C) and *myc2* (*myc2*-1, SALK_061267C; *myc2*-2, SALK_128938C) mutants from the Arabidopsis Biological Resource Center (ABRC) [83]. The *ataf1* mutant has a T-DNA insertion in the third exon of the *ATAF1* (AT1G01720) genomic DNA, both *myc2* mutants have a T-DNA insertion in the exon of the *MYC2* (AT1G32640) genomic DNA (Fig 8A). We germinated wild-type (WT) Col-0 and *ataf1* mutant on the half-strength Murashige and Skoog (MS) medium with or without 300 mM mannitol treatment (Fig 8B). The addition of mannitol reduces water potential of growth media, which is often used to mimic drought stress (Mu et al., 2019) [84]. Although the germination rate of the *ataf1* mutant was lower than WT in the medium without mannitol, the *ataf1* mutant had more green cotyledon seedlings (Fig 8B) and higher green cotyledon rate (Fig 8C) than WT seedlings under 300 mM mannitol treatment. The difference became significant at nine days after germination. We also compared the green cotyledon inhibition rate of WT and *ataf1* mutant on MS medium with or without mannitol. Consistently, the *ataf1* mutant showed lower green cotyledon inhibition rate than WT, and the tendency became more pronounced with the increase of growth time (Fig 8D). We also germinated WT and *myc2* mutants on the MS medium with or without 300

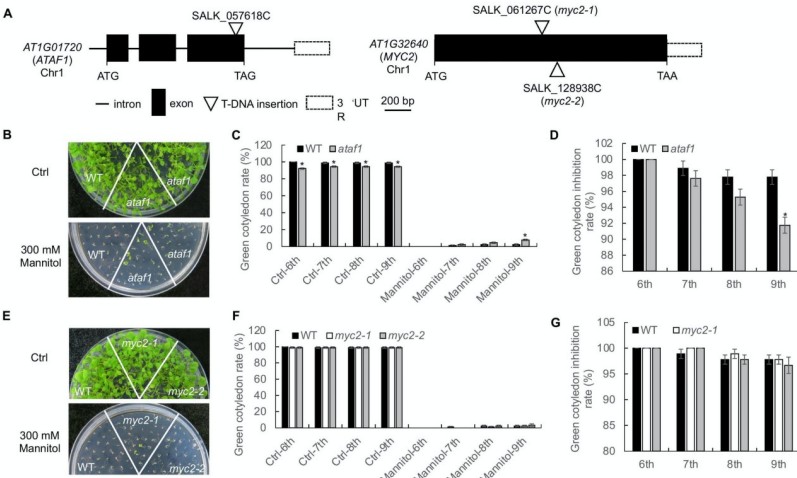

**Fig 8. Results from validation experiments. A**. The scheme of the *ATAF1* and *MYC2* genomic DNA and T-DNA insertion. The panel is a schematic illustration of the *ATAF1* and *MYC2* genomic DNA with exons (solid box), intron (lines) and 3' untranslated region (open box). The position of T-DNA insertion of *ataf1* (SALK_057618C), *myc2* (SALK_061267C, SALK_128938C)was labeled. **B**. The *ataf1* mutant is more resistant to mannitol treatment. Wild-type (WT) Col-0 and *ataf1* mutant seeds were germinated on 1/2 MS medium with or without 300 mM mannitol. 30 seeds per genotype were used for each replicate. The photos were taken four-week post-germination. **C**. Quantification of cotyledon greening on plates corresponding to B. Seedlings with green cotyledon expansion were counted at 6–9 days post-germination. Data are shown as means ± SD (standard deviation) from three independent replicates (n = 3, *, p<0.05, Student's t-test). **D**. Quantification of cotyledon greening inhibition rate on plates corresponding to B. Seedlings with green cotyledon expansion were counted at 6–9 days post-germination. Data are shown as means ± SD from three independent replicates (n = 3, *, p<0.05, Student's t-test). **E**. Growth of WT and *myc2* mutants on MS plates. WT and *myc2* mutant seeds were germinated on 1/2 MS medium with or without 300 mM mannitol. 30 seeds per genotype were used for each replicate. The photos were taken four-week post-germination. **F**. Quantification of cotyledon greening on plates corresponding to E. Seedlings with green cotyledon expansion were counted at 6–9 days post-germination. Data are shown as means ± SD from three independent replicates (n = 3, no statistical significance with Student's t-test). **G**. Quantification of cotyledon greening on plates corresponding to E. Seedlings with green cotyledon expansion were counted at 6–9 days post-germination. Data are shown as means ± SD from three independent replicates (n = 3, no statistical significance with Student's t-test).

mM mannitol treatment (Fig 8E). However, there is no significant difference in the green cotyledon rate between WT and *myc2* mutants with or without mannitol treatment (Fig 8F). Similarly, the green cotyledon inhibition rate between WT and *myc2* mutants also did not show a significant difference (Fig 8G). Thus, our data show that the *ataf1* mutant was more tolerant to the mannitol treatment, and suggests that *ATAF1* plays a role in plant drought stress response. Our test conditions, such as plant growth stage, treatment, or the combination, may not be suitable to reveal the difference between WT and *myc2* mutants.

## Experimental setup

A. thaliana mutants *ataf1* (SALK_057618C) and *myc2* (SALK_061267C, SALK_128938C) were obtained from the Arabidopsis Biological Resource Center (ABRC). The wild-type (Col-0) and mutant plants were grown in a growth room at 23˚C, 45% humidity, and 75 $\mu$E m$^{-2}$ s$^{-1}$ light with a 12-hr light /12-hr dark photoperiod. To detect cotyledon greening rate, 30 seeds per genotype were sterilized and germinated on half-strength Murashige and Skoog (MS) medium with or without 300 mM Mannitol treatment in each replicate. Seedlings with green cotyledon expansion were counted at 6–9 d post-germination, data are shown as means ± SD from three independent repeats (n = 3, *, p<0.05, Student's t-test). The photos were taken four-weeks post-germination.

## Discussion

As the severity and duration of droughts around the world are predicted to rise in the coming years, developing drought resistant crops is increasingly becoming a priority for ensuring global food security. Thus to develop drought resistant crops, it is necessary for scientists to identify the potent regulators of the drought response in plants. In this paper, we have presented the drought signaling pathway in Arabidopsis and observed that drought response is mediated by the ABA dependent or several ABA-independent pathways. We selected the model plant Arabidopsis for our study because the genes and proteins in drought response pathways are well defined and identified for Arabidopsis compared to major crops. We modeled these pathways using BNs, as it provides a framework to integrate both biological prior knowledge in the form of pathway information along with experimental data. This feature of BNs was a key factor in our selection of this modeling technique. In the BN model, we assumed each node to be a binary random variable with the states of activation or inhibition. We then used the Bayesian approach along with publicly available experimental data to estimate the LPDs associated with the nodes of the BN model. The prior distribution for each node was assumed to follow a Beta(1,1) distribution as this corresponds to the non-informative Uniform distribution on the interval [0, 1]. This choice of prior was logical as we did not know the prior distribution for each of the nodes. Furthermore, choosing a Beta prior with Binomial likelihood provides us with a closed form solution for the posterior distribution and reduces our computational requirements. Once the LPDs were learned, we applied an approximate inference technique called likelihood weighting to perform simulations for intervening at the non-reporter gene nodes.

After intervening at the nodes representing the non-reporter genes, one at a time, we observed that the scores were maximized upon activating *MYC2* or inhibiting *ATAF1*. The maximization of scores implied that *MYC2* and *ATAF1* were potential drought regulators, and activating *MYC2* or inhibiting *ATAF1* was the best strategy to regulate the drought-responsive reporter genes. We also observed that the score for implementing both these interventions at the same time provides a slightly improved score value, indicating the synergistic effect of the strategic interventions. These simulation results indicated that *ATAF1* and *MYC2* were the most potent regulators of drought response compared to the other drought regulatory genes modeled in the BN.

From biological literature we note that both *MYC2* and *ATAF1* are known regulators of drought response. However, from the validation experiments, we found that *MYC2* did not have any obvious drought regulatory response as neither the green cotyledon rate nor the green cotyledon inhibition rate between WT and *myc2* mutants with or without mannitol treatment had significant differences. On the other hand, *ataf1* mutants had more green cotyledon seedlings and higher green cotyledon rates than the WT seedlings under mannitol treatment, suggesting that *ATAF1* negatively regulated drought response. We were unable to show that *MYC2* was a drought regulator; this could be due to test conditions or limitations of the Bayesian network model. Testing factors such as plant growth stage, treatment may have been unfavorable for finding the difference between WT and *myc2* mutants. Besides testing factors, we must also consider some of the limitations of the BN model. While we have considered numerous drought-responsive pathways in our BN model, there may be other pathways outside our model's scope, which may interact with the pathways considered in our BN model. These undiscovered interactions may have potentially influenced the drought regulators during the validation experiments. In order to avoid neglecting such interactions, BNs are learned from data using structure learning algorithms. However, this process typically requires large volumes of data, which is currently unavailable. Furthermore, if any previously unaccounted

interactions are discovered using structure learning algorithm, we cannot validate them using existing biological literature, and we will need to conduct additional experiments to validate them. Another reason that might have prevented us from proving *MYC2* as a drought regulator is the difference between the experimental setup of our validation experiments and the publicly available dataset(GSE42408) used to learn the parameters of the BN model. The methods used to induce drought in the dataset GSE42408 are different from the methods used in our validation experiments; this might have been unfavorable in establishing *MYC2* as drought regulator.

This paper's results build upon our previous paper, where we modeled only the WRKY transcription factor signaling pathway in Arabidopsis under drought and found the transcription factor *WRKY18* to be the best regulator of the drought-responsive gene *RD29A* [49]. In our current model, we take into account multiple other pathways, including the WRKY signaling pathway, and observe that the scores across the WRKY transcription factor family are approximately the same and are not as high as the scores for *MYC2* and *ATAF1*. The score for *WRKY18* may be low due to crosstalk happening across multiple pathways, which may negatively impact the regulatory effects of *WRKY18*. Additionally, we tracked multiple drought-responsive reporter genes in our current study, so the score of *WRKY18* in this study reflects its ability to regulate all the drought-responsive reporter genes, unlike in the previous paper, where the score is for the regulation of *RD29A* only. In the future, we would like to extend our research to include more informative priors instead of the non-informative Beta (1,1) distribution. We want to explore new methods to incorporate continuous data into the BN model, rather than to binarize it and lose valuable information. We noticed that multi-node intervention gave a slightly improved score than single node interventions; thus, exploring other node combinations for intervention will be an interesting path for future research.

## Conclusion

We modeled several drought-responsive pathways in Arabidopsis using Bayesian Networks and real-world experimental data. Our computational analysis indicated that the transcription factors *MYC2* and *ATAF1* are the most potent candidates for regulating drought-responsive reporter genes. However, we were only able to validate the drought regulatory response of *ATAF1* experimentally. Since *ATAF1* had the highest score for inhibition and validation experiments showed all *ataf1* mutants had a higher green cotyledon rate than WT, it implies that *ATAF1* negatively regulates drought response. Thus genetically inhibiting *ATAF1* with techniques such as CRISPR-Cas9 has the potential to develop drought-resistant crops.

## Supporting information

**S1 File. Main R code file for executing the Bayesian network simulation.**
(R)

**S2 File. Supporting R code file for normalizing the data.**
(R)

**S3 File. Supporting R code file for binarizing the data.**
(R)

**S4 File. Supporting R code file for calculating shape parameters.**
(R)

**S5 File. Supporting R code file for renaming the dataset with appropriate gene names.**
(R)

**S6 File. This file contains the subset of the dataset GSE42408, which supports the conclusion of this article.** This subset includes the data under drought conditions for pertinent genes involved in the Bayesian network analysis. The complete dataset can be publicly accessed online from the NCBI GEO database with the accession number of GSE42408.
(CSV)

## Author Contributions

**Conceptualization:** Aditya Lahiri, Ping He, Aniruddha Datta.

**Data curation:** Aditya Lahiri.

**Formal analysis:** Aditya Lahiri, Aniruddha Datta.

**Funding acquisition:** Ping He, Aniruddha Datta.

**Investigation:** Aditya Lahiri, Ping He, Aniruddha Datta.

**Methodology:** Aditya Lahiri, Lin Zhou.

**Project administration:** Ping He, Aniruddha Datta.

**Resources:** Ping He, Aniruddha Datta.

**Software:** Aditya Lahiri.

**Supervision:** Ping He, Aniruddha Datta.

**Validation:** Lin Zhou, Ping He.

**Visualization:** Aditya Lahiri, Lin Zhou.

**Writing – original draft:** Aditya Lahiri, Lin Zhou.

**Writing – review & editing:** Aditya Lahiri, Lin Zhou, Ping He, Aniruddha Datta.

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
