## [Decision Letter · Decision Letter 0]

17 Jun 2021

PONE-D-21-16503

Detecting Drought Regulators using Stochastic Inference in Bayesian Networks

PLOS ONE

Dear Dr. Lahiri,

Thank you for submitting your manuscript to PLOS ONE. After careful consideration, we feel that it has merit but does not fully meet PLOS ONE’s publication criteria as it currently stands. Therefore, we invite you to submit a revised version of the manuscript that addresses the points raised during the review process. The reviewers have highlighted a few relatively minor points that should help to clarify aspects of the paper. Once addressed, the paper will be acceptable for publication in PLOS ONE.

We look forward to receiving your revised manuscript.

Kind regards,

Steven M. Abel, Ph.D.

Academic Editor

PLOS ONE

Journal Requirements:

Reviewers' comments:

Reviewer's Responses to Questions

**Comments to the Author**

1. Is the manuscript technically sound, and do the data support the conclusions?

Reviewer #1: Yes

Reviewer #2: Yes

2. Has the statistical analysis been performed appropriately and rigorously? 

Reviewer #1: Yes

Reviewer #2: Yes

3. Have the authors made all data underlying the findings in their manuscript fully available?

Reviewer #1: Yes

Reviewer #2: Yes

4. Is the manuscript presented in an intelligible fashion and written in standard English?

Reviewer #1: No

Reviewer #2: Yes

5. Review Comments to the Author

Reviewer #1: This paper presents and interesting framework to study drought responses in the context of gene regulatory networks. The authors did a good job discussing the caveats relevant for interpreting their results.

Regarding question 4 - the writing is overall fine but I have concerns about the explanation of some analyses.

Lines 353-359 are a difficult to follow description of the analysis of expression data:

The authors describe their raw data vaguely as “eQTL data points” but do not define what this means in precise terms. It was necessary to read the paper from which the authors obtained their expression data and manually read through their R scripts to understand the analyses. For example, the authors could state that they analyzed expression data of the genes of interest under drought conditions in 104 recombinant inbred lines.

Reviewer #2: This study used prior information on four drought reporter genes and their regulators in Arabidopsis, and applied Bayesian inference technique to study the effect of perturbation to regulators on the reporter genes. The study draws inference on the activity of network nodes using experimental data, and the algorithm that simulates node intervention is well described. The study also provides experimental verification of the model, and some of it could be validated experimentally. Overall, the study seems to derive a potential mechanism underlying the known drought regulatory network, rather than finding new regulators of drought. As such, I see this as a the only limitation of the study, as the nodes in the BN model are already well characterized in Arabidopsis but do not easily translate to major crops. It would be nice to state this in the discussion.

What was the rationale behind using eQTL data for LPD estimations? Could RNA-seq data be used?

Minor:

Line 10: California is duplicated. Please rephrase.

Line 191: …., the local marginal probability distribution is given by θICE1? Will the cpt of ice not dependent on all its parent nodes?

Line: 468: “...that activating MYC2 or inhibiting ATAF1 468 was the best strategy to regulate the drought-responsive reporter genes”. Explain why it is the best?

6. PLOS authors have the option to publish the peer review history of their article (what does this mean?). If published, this will include your full peer review and any attached files.

Reviewer #1: No

Reviewer #2: **Yes: **Chirag Gupta

---

## [Author Response · Author response to Decision Letter 0]

15 Jul 2021

July 14, 2021

Dear Reviewers,

We would like to thank you for taking the time and effort to review our manuscript and code. We sincerely appreciate all the constructive feedback and suggestions you have provided towards improving our manuscript. We have first addressed the comments of Reveiwer#1 and then Reviewer#2. The responses to Reviewer comments are made in a point-by-point fashion below. Reviewer comments and PLOS ONE requirements are presented in italics. The changes listed below have been incorporated and highlighted in our revised manuscript. All the Authors have reviewed and approved these changes.

Response to Reviewer #1

1. Is the manuscript technically sound, and do the data support the conclusions?

Reviewer #1: Yes

Author Response: We thank the Reviewer for their evaluation.

2. Has the statistical analysis been performed appropriately and rigorously?

Reviewer #1: Yes

Author Response: We thank the Reviewer for their evaluation.

3. Have the authors made all data underlying the findings in their manuscript fully available?

Reviewer #1: Yes

Author Response: We thank the Reviewer for their evaluation.

4. Is the manuscript presented in an intelligible fashion and written in standard English?

Reviewer #1: No

Author Response: We have made the changes listed below (section and line wise) to improve the language, presentation, and clarity of our manuscript. 

Abstract: 

Line 2: Removed comma from the second sentence after the word “drought”.

Introduction: 

Line 5: Added comma after the word “deficit”.

Line 14: Added comma after the word “China”. 

Line 27: Replaced the word “which” with “that”.

Line 28-30: Restructured the sentence to make it easier to read.

Line 35: Changed wording to “plant breeding methods”.

Plant Defense Mechanism:

Line 58-59: Added comma after the words “nitrogen” and “acid”.

Line 68: Added comma after the words “drought” and “avoidance”.

Line 74: Added comma after the word “instead”.

Drought Signaling Networks:

Line 87: Remove comma after the word “pathways”.

Line 92-95: Restructured sentence to improve the explanation of abscisic acid. 

Line 101-103: Restructured sentence to clarify the role of abscisic and jasmonic acid in drought regulation. 

Line 111: Added comma after the word “factor”. 

Line 116: Added comma after the word “here”, changed the word “independent” to “independently”.

Line 125: Removed comma after the word “2017”.

Line 130: Added comma after the word “Arabidopsis”.

Line 137: Added comma after the word “RD29A”.

Line 145-146: Improved the sentence structure to clarify the role of MYB2 and MYC2 in the regulation of ERD1.

Bayesian Network Model:

Line 179: Added comma after the word “protein”

Line 181-182: Rephrased the sentence to better explain the binary nature of the nodes in the BN. Line 191-194: Rephrased the sentences to better explain the workflow of BNs from LPD estimation to inference.

Parameter Estimation in Bayesian Networks:

Line 200-201: Clarified reference to prior sections and added comma after the word “defined”.

Line 218-220: Restructured the sentence to clarify the drawback involving prior distribution in BN.

Sampling based Inference in Bayesian Networks:

Line 262-264: Added a comma after the word “section”. Removed comma after the word “model”. Changed “to determine” to “in determining”.

Line 266: Removed comma after the word “time” and added comma after word “1”

Line 273-278: Rephrased these lines to clarify the concept of multiply connected networks and the limitation of exact inference. 

Line 281: Deleted the word “would” after the word “BN”.

Line 319: Added comma after the word “variables”

Line 325: Added comma after the word “regulated”

Results:

Line 371: Removed comma after the word “GSE42408”

Line 376: Removed comma after the word “ERD1”

Line 379: Added comma after the word “analysis”

Line 387: Deleted the word “In” before the word “Figs 5”

Line 389: Added commas after the words “Fig 6” and “Fig 5”.

Line 391: Added comma after the word “scores”.

Line 392: Added comma after the word “inhibition”

Line 392-395: Restructured the sentences better to explain the results of single node intervention analysis. 

Line 398: Removed comma after the word “analysis”

Line 406-407: Reworded the sentence to include more specific detail on the cited study.

Line 411: Removed comma after the word “genes”.

Line 413: Added comma after the word “inhibition”.

Additional Changes:

References to figures in the manuscript have been changed Fig.# to Fig # to meet the style requirements of PLOS ONE.

Line 157: Capitalized the alphabet “M” in the section heading “Materials and Methods”.

Line 195: Capitalized the initial alphabets of the words “estimation” and “networks” in the section heading “Parameter Estimation in Bayesian Networks”.

Line 261: Capitalized the initial alphabets of the words “inference” and “networks” in the section heading “Sampling Based Inference in Bayesian Networks”.

Line 349-350: Fixed the position of Table 1.

5. Review Comments to the Author

Reviewer #1 Comment #1: This paper presents and interesting framework to study drought responses in the context of gene regulatory networks. The authors did a good job discussing the caveats relevant for interpreting their results.

Regarding question 4 - the writing is overall fine but I have concerns about the explanation of some analyses.

Author Response: We thank the Reviewer for their kind words. We have made the following changes to improve the explanation of various analysis in our paper. 

Lines 28-30: Restructured the sentence to make it easier to read and better introduce the notion of the internal drought defensive mechanism of plants. 

Line 92-95: Restructured sentence to improve the explanation of abscisic acid.

Line 101-103: Restructured sentence to clarify the role of abscisic and jasmonic acid in drought regulation.

Line 145-146: Improved the sentence structure to clarify the role of MYB2 and MYC2 in the regulation of ERD1.

Line 181-182: Rephrased the sentence to better explain the binary nature of the nodes in the BN.

Line 191-194: Rephrased the sentences to better explain the workflow of BNs from LPD estimation to inference. 

Line 218-220: Restructured the sentence to clarify the drawback involving prior distribution in BN.

Line 273-278: Rephrased these lines to clarify the concept of multiply connected networks and the limitation of exact inference.

Line 392-395: Restructured the sentences better to explain the results of the single node intervention analysis.

Line 406-407: Reworded the sentence to include more specific detail on the cited study.

Reviewer #1 Comment #2: Lines 353-359 are difficult to follow description of the analysis of expression data: The authors describe their raw data vaguely as “eQTL data points” but do not define what this means in precise terms. It was necessary to read the paper from which the authors obtained their expression data and manually read through their R scripts to understand the analyses. For example, the authors could state that they analyzed expression data of the genes of interest under drought conditions in 104 recombinant inbred lines.

Author Response: We appreciate the Reviewer for raising this concern and for their helpful suggestions. We have incorporated these suggestions in Lines 351-357. Additionally, we have added the rationale behind choosing this dataset.

Response to Reviewer #2

1. Is the manuscript technically sound, and do the data support the conclusions?

Reviewer #2: Yes

Author Response: We thank the Reviewer for their evaluation.

2. Has the statistical analysis been performed appropriately and rigorously?

Reviewer #2: Yes

Author Response: We thank the Reviewer for their evaluation.

3. Have the authors made all data underlying the findings in their manuscript fully available?

Reviewer #2: Yes

Author Response: We thank the Reviewer for their evaluation.

4. Is the manuscript presented in an intelligible fashion and written in standard English?

Reviewer #2: Yes

Author Response: We thank the Reviewer for their evaluation.

5. Review Comments to the Author

Reviewer #2 Comment #1: This study used prior information on four drought reporter genes and their regulators in Arabidopsis, and applied Bayesian inference technique to study the effect of perturbation to regulators on the reporter genes. The study draws inference on the activity of network nodes using experimental data, and the algorithm that simulates node intervention is well described. The study also provides experimental verification of the model, and some of it could be validated experimentally. Overall, the study seems to derive a potential mechanism underlying the known drought regulatory network, rather than finding new regulators of drought. As such, I see this as a the only limitation of the study, as the nodes in the BN model are already well characterized in Arabidopsis but do not easily translate to major crops. It would be nice to state this in the discussion.

Author Response: We thank the Reviewer for their insightful comments. We have incorporated the requested changes in the discussion section in Lines 457-459.

Reviewer #2 Comment #2: What was the rationale behind using eQTL data for LPD estimations? Could RNA-seq data be used?

Author Response: We thank the Reviewer for raising this question. We have explained the rationale of using eQTL data and other details pertaining to this specific dataset in Lines 351-358. Yes, we can use RNA-seq data as well for the estimations of LPDs.

Reviewer #2 Comment #3: Minor: Line 10: California is duplicated. Please rephrase.

Author Response: We thank the Reviewer for notifying us about this typographical error, we have remedied this error in Line 10-11.

Reviewer #2 Comment #4: Line 191: …., the local marginal probability distribution is given by θICE1? Will the cpt of ice not dependent on all its parent nodes?

Author Response: We thank the Reviewer for notifying this error. Indeed, the CPT on a node is dependent on all its parent nodes, and ICE1 is dependent on HOS1 and SIZ1. The corrections are incorporated in Lines 188-190.

Reviewer #2 Comment #5: Line: 468: “...that activating MYC2 or inhibiting ATAF1 468 was the best strategy to regulate the drought-responsive reporter genes”. Explain why it is the best?

Author Response: We thank the Reviewer for raising this concern. We have provided the explanation in Lines 473-477.

---

## [Editor Report · Decision Letter 1]

19 Jul 2021

Detecting Drought Regulators using Stochastic Inference in Bayesian Networks

PONE-D-21-16503R1

Dear Dr. Lahiri,

We’re pleased to inform you that your manuscript has been judged scientifically suitable for publication and will be formally accepted for publication once it meets all outstanding technical requirements.

Kind regards,

Steven M. Abel, Ph.D.

Academic Editor

PLOS ONE

---

## [Editor Report · Acceptance letter]

6 Aug 2021

PONE-D-21-16503R1 

Detecting Drought Regulators using Stochastic Inference in Bayesian Networks 

Dear Dr. Lahiri:

I'm pleased to inform you that your manuscript has been deemed suitable for publication in PLOS ONE. Congratulations! Your manuscript is now with our production department. 

Kind regards, 

on behalf of

Dr. Steven M. Abel 

Academic Editor

PLOS ONE